# End-to-end named entity recognition and relation extraction using pre-trained language models

## Abstract

Named entity recognition (NER) and relation extraction (RE) are two important tasks in information extraction and retrieval (IE & IR). Recent work has demonstrated that it is beneficial to learn these tasks jointly, which avoids the propagation of error inherent in pipeline-based systems and improves performance. However, state-of-the-art joint models typically rely on external natural language processing (NLP) tools, such as dependency parsers, limiting their usefulness to domains (e.g. news) where those tools perform well. The few neural, end-to-end models that have been proposed are trained almost completely from scratch. In this paper, we propose a neural, end-to-end model for jointly extracting entities and their relations which does not rely on external NLP tools and which integrates a large, pre-trained language model. Because the bulk of our model's parameters are pre-trained and we eschew recurrence for self-attention, our model is fast to train. On 5 datasets across 3 domains, our model matches or exceeds state-of-the-art performance, sometimes by a large margin.

## 1 Introduction

The extraction of named entities (named entity recognition, NER) and their semantic relations (relation extraction, RE) are key tasks in information extraction and retrieval (IE & IR). Given a sequence of text (usually a sentence), the objective is to identify both the named entities and the relations between them. This information is useful in a variety of NLP tasks such as question answering, knowledge base population, and semantic search (Jiang, 2012). In the biomedical domain, NER and RE facilitate large-scale biomedical data analysis, such as network biology (Zhou et al., 2014), gene prioritization (Aerts et al., 2006), drug repositioning (Wang & Zhang, 2013) and the creation of curated databases (Li et al., 2015). In the clinical domain, NER and RE can aid in disease and treatment prediction, readmission prediction, de-identification, and patient cohort identification (Miotto et al., 2017).

Most commonly, the tasks of NER and RE are approached as a pipeline, with NER preceding RE. There are two main drawbacks to this approach: (1) Pipeline systems are prone to error propagation between the NER and RE systems. (2) One task is not able to exploit useful information from the other (e.g. the *type* of relation identified by the RE system may be useful to the NER system for determining the *type* of entities involved in the relation, and vice versa). More recently, joint models that simultaneously learn to extract entities and relations have been proposed, alleviating the aforementioned issues and achieving state-of-the-art performance (Miwa & Sasaki, 2014; Miwa & Bansal, 2016; Gupta et al., 2016; Li et al., 2016; 2017; Zhang et al., 2017; Adel & Schütze, 2017; Bekoulis et al., 2018a;b; Nguyen & Verspoor, 2019; Li et al., 2019).

Many of the proposed joint models for entity and relation extraction rely heavily on external natural language processing (NLP) tools such as dependency parsers. For instance, Miwa & Bansal (2016) propose a recurrent neural network (RNN)-based joint model that uses a bidirectional long-short term memory network (BiLSTM) to model the entities and a tree-LSTM to model the relations between entities; Li et al. (2017) propose a similar model for biomedical text. The tree-LSTM uses dependency tree information extracted using an external dependency parser to model relations between entities. The use of these external NLP tools limits the effectiveness of a model to domains

(e.g. news) where those NLP tools perform well. As a remedy to this problem, Bekoulis et al. (2018a) proposes a neural, end-to-end system that jointly learns to extract entities and relations without relying on external NLP tools. In Bekoulis et al. (2018b), they augment this model with adversarial training. Nguyen & Verspoor (2019) propose a different, albeit similar end-to-end neural model which makes use of deep biaffine attention (Dozat & Manning, 2016). Li et al. (2019) approach the problem with multi-turn question answering, posing templated queries to a BERT-based QA model (Devlin et al., 2018) whose answers constitute extracted entities and their relations and achieve state-of-the-art results on three popular benchmark datasets.

While demonstrating strong performance, end-to-end systems like Bekoulis et al. (2018a;b) and Nguyen & Verspoor (2019) suffer from two main drawbacks. The first is that most of the models parameters are trained from scratch. For large datasets, this can lead to long training times. For small datasets, which are common in the biomedical and clinical domains where it is particularly challenging to acquire labelled data, this can lead to poor performance and/or overfitting. The second is that these systems typically contain RNNs, which are sequential in nature and cannot be parallelized within training examples. The multi-pass QA model proposed in Li et al. (2019) alleviates these issues by incorporating a pre-trained language model, BERT (Devlin et al., 2018), which eschews recurrence for self-attention. The main limitation of their approach is that it relies on hand-crafted question templates to achieve maximum performance. This may become a limiting factor where domain expertise is required to craft such questions (e.g., for biomedical or clinical corpora). Additionally, one has to create a question template for each entity and relation type of interest.

In this study, we propose an end-to-end model for joint NER and RE which addresses all of these issues. Similar to past work, our model can be viewed as a mixture of a NER module and a RE module (Figure 1). Unlike most previous works, we include a pre-trained, transformer-based language model, specifically BERT (Devlin et al., 2018), which achieved state-of-the-art performance across many NLP tasks. The weights of the BERT model are fine-tuned during training, and the entire model is trained in an end-to-end fashion.

Our main contributions are as follows: (1) Our solution is truly end-to-end, relying on no hand-crafted features (e.g. templated questions) or external NLP tools (e.g. dependency parsers). (2) Our model is fast to train (e.g. under 10 minutes on a single GPU for the CoNLL04 corpus), as most of its parameters are pre-trained and we avoid recurrence. (3) We match or exceed state-of-the-art performance for joint NER and RE on 5 datasets across 3 domains.

## 2 THE MODEL

Figure 1 illustrates the architecture of our approach. Our model is composed of an NER module and an RE module. The NER module is identical to the one proposed by Devlin et al. (2018). For a given input sequence $s$ of $N$ word tokens $w_1, w_2, \ldots, w_N$, the pre-trained $\text{BERT}_{\text{BASE}}$ model first produces a sequence of vectors, $\boldsymbol{x}_1^{(\text{NER})}, \boldsymbol{x}_2^{(\text{NER})}, \ldots, \boldsymbol{x}_N^{(\text{NER})}$ which are then fed to a feed-forward neural network (FFNN) for classification.

$$\boldsymbol{s}_i^{(\text{NER})} = \text{FFNN}_{\text{NER}}(\boldsymbol{x}_i^{(\text{NER})}) \tag{1}$$

The output size of this layer is the number of BIOES-based NER labels in the training data, $|C^{(\text{NER})}|$. In the BIOES tag scheme, each token is assigned a label, where the B- tag indicates the beginning of an entity span, I- the inside, E- the end and S- is used for any single-token entity. All other tokens are assigned the label O.

During training, a cross-entropy loss is computed for the NER objective,

$$\mathcal{L}_{\text{NER}} = -\sum_{n=1}^{N} \log \left( \frac{e^{\boldsymbol{s}_n^{(\text{NER})}}}{\sum_c^{C^{(\text{NER})}} e^{\boldsymbol{s}_{n,c}^{(\text{NER})}}} \right) \tag{2}$$

where $\boldsymbol{s}_n^{(\text{NER})}$ is the predicted score that token $n \in N$ belongs to the ground-truth entity class and $\boldsymbol{s}_{n,c}^{(\text{NER})}$ is the predicted score for token $n$ belonging to the entity class $c \in C^{(\text{NER})}$.

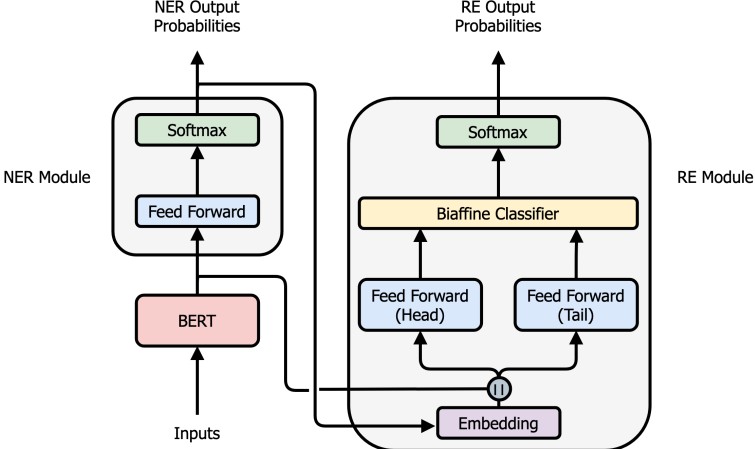

Figure 1: Joint named entity recognition (NER) and relation extraction (RE) model architecture.

In the RE module, the predicted entity labels are obtained by taking the argmax of each score vector $s_1^{(\text{NER})}$, $s_2^{(\text{NER})}$, ..., $s_N^{(\text{NER})}$. The predicted entity labels are then embedded to produce a sequence of fixed-length, continuous vectors, $e_1^{(\text{NER})}$, $e_2^{(\text{NER})}$, ..., $e_N^{(\text{NER})}$ which are concatenated with the hidden states from the final layer in the BERT model and learned jointly with the rest of the models parameters.

$$x_i^{(\text{RE})} = x_i^{(\text{NER})} \parallel e_i^{(\text{NER})} \tag{3}$$

Following Miwa & Bansal (2016) and Nguyen & Verspoor (2019), we incrementally construct the set of relation candidates, $R$, using all possible combinations of the last word tokens of predicted entities, i.e. words with E- or S- labels. An entity pair is assigned to a negative relation class (NEG) when the pair has no relation or when the predicted entities are not correct. Once relation candidates are constructed, classification is performed with a deep bilinear attention mechanism (Dozat & Manning, 2016), as proposed by Nguyen & Verspoor (2019).

To encode directionality, the mechanism uses FFNNs to project each $x_i^{(\text{RE})}$ into *head* and *tail* vector representations, corresponding to whether the $i^{th}$ word serves as head or tail argument of the relation.

$$h_i^{(\text{head})} = \text{FFNN}_{\text{head}}(x_i^{(\text{RE})}) \tag{4}$$

$$h_i^{(\text{tail})} = \text{FFNN}_{\text{tail}}(x_i^{(\text{RE})}) \tag{5}$$

These projections are then fed to a biaffine classifier,

$$s_{j,k}^{(\text{RE})} = \text{Biaffine}(h_j^{(\text{head})}, h_k^{(\text{tail})}) \tag{6}$$

$$\text{Biaffine}(x_1, x_2) = x_1^T \mathbf{U} x_2 + W(x_1 \parallel x_2) + b \tag{7}$$

where $\mathbf{U}$ is an $m \times |C^{(\text{RE})}| \times m$ tensor, $W$ is a $|C^{(\text{RE})}| \times (2 * m)$ matrix, and $b$ is a bias vector. Here, $m$ is the size of the output layers of $\text{FFNN}_{\text{head}}$ and $\text{FFNN}_{\text{tail}}$ and $C^{(\text{RE})}$ is the set of all relation classes (including NEG). During training, a second cross-entropy loss is computed for the RE objective

$$\mathcal{L}_{\text{RE}} = -\sum_{r=1}^{R} \log \left( \frac{e^{s_r^{(\text{RE})}}}{\sum_c^{C^{(\text{RE})}} e^{s_{r,c}^{(\text{RE})}}} \right) \tag{8}$$

where $s_r^{(\text{RE})}$ is the predicted score that relation candidate $r \in R$ belongs to the ground-truth relation class and $s_{r,c}^{(\text{RE})}$ is the predicted score for relation $r$ belonging to the relation class $c \in C^{(\text{RE})}$.

The model is trained in an end-to-end fashion to minimize the sum of the NER and RE losses.

$$\mathcal{L} = \mathcal{L}_{\text{NER}} + \mathcal{L}_{\text{RE}} \qquad (9)$$

## 2.1 Entity Pretraining

In Miwa & Bansal (2016), entity pre-training is proposed as a solution to the problem of low-performance entity detection in the early stages of training. It is implemented by delaying the training of the RE module by some number of epochs, before training the entire model jointly.

Our implementation of entity pretraining is slightly different. Instead of delaying training of the RE module by some number of epochs, we weight the contribution of $\mathcal{L}_{\text{RE}}$ to the total loss during the first epoch of training

$$\mathcal{L} = \mathcal{L}_{\text{NER}} + \lambda \, \mathcal{L}_{\text{RE}} \qquad (10)$$

where $\lambda$ is increased linearly from 0 to 1 during the first epoch and set to 1 for the remaining epochs. We chose this scheme because the NER module quickly achieves good performance for all datasets (i.e. within one epoch). In early experiments, we found this scheme to outperform a delay of a full epoch.

## 2.2 Implementation

We implemented our model in PyTorch (Paszke et al., 2017) using the BERT$_{\text{BASE}}$ model from the PyTorch Transformers library[1]. Our model is available at our GitHub repository[2]. Furthermore, we use NVIDIAs automatic mixed precision (AMP) library Apex[3] to speed up training and reduce memory usage without affecting task-specific performance.

## 3 Experimental Setup

### 3.1 Datasets and evaluation

To demonstrate the generalizability of our model, we evaluate it on 5 commonly used benchmark corpora across 3 domains. All corpora are in English. Detailed corpus statistics are presented in Table A.1 of the appendix.

### 3.1.1 ACE04/05

The Automatic Content Extraction (ACE04) corpus was introduced by Doddington et al. (2004), and is commonly used to benchmark NER and RE methods. There are 7 entity types and 7 relation types. ACE05 builds on ACE04, splitting the *Physical* relation into two classes (*Physical* and *Part-Whole*), removing the *Discourse* relation class and merging *Employment-Membership-Subsidiary* and *Person-Organization-Affiliation* into one class (*Employment-Membership-Subsidiary*).

For ACE04, we follow Miwa & Bansal (2016) by removing the *Discourse* relation and evaluating our model using 5-fold cross-validation on the *bnews* and *nwire* subsets, where 10% of the data was held out within each fold as a validation set. For ACE05, we use the same test split as Miwa & Bansal (2016). We use 5-fold cross-validation on the remaining data to choose the hyperparameters. Once hyperparameters are chosen, we train on the combined data from all the folds and evaluate on the test set. For both corpora, we report the micro-averaged F$_1$ score. We obtained the pre-processing scripts from Miwa & Bansal (2016)[4].

---

[1]https://github.com/huggingface/pytorch-transformers
[2]to-add-to-camera-ready-version
[3]https://github.com/NVIDIA/apex
[4]https://github.com/tticoin/LSTM-ER/tree/master/data

### 3.1.2 CoNLL04

The CoNLL04 corpus was introduced in Roth & Yih (2004) and consists of articles from the Wall Street Journal (WSJ) and Associated Press (AP). There are 4 entity types and 5 relation types.

We use the same test set split as Miwa & Sasaki (2014)[5]. We use 5-fold cross-validation on the remaining data to choose hyperparameters. Once hyperparameters are chosen, we train on the combined data from all folds and evaluate on the test set, reporting the micro-averaged $F_1$ score.

### 3.1.3 ADE

The adverse drug event corpus was introduced by Gurulingappa et al. (2012) to serve as a benchmark for systems that aim to identify adverse drug events from free-text. It consists of the abstracts of medical case reports retrieved from PubMed[6]. There are two entity types, *Drug* and *Adverse effect* and one relation type, *Adverse drug event*.

Similar to previous work (Li et al., 2016; 2017; Bekoulis et al., 2018b), we remove ∼130 relations with overlapping entities and evaluate our model using 10-fold cross-validation, where 10% of the data within each fold was used as a validation set, 10% as a test set and the remaining data is used as a train set. We report the macro $F_1$ score averaged across all folds.

### 3.1.4 I2B2

The 2010 i2b2/VA dataset was introduced by Uzuner et al. (2011) for the 2010 i2b2/Va Workshop on Natural Language Processing Challenges for Clinical Records. The workshop contained an NER task focused on the extraction of 3 medical entity types (*Problem*, *Treatment*, *Test*) and an RE task for 8 relation types.

In the official splits, the test set contains roughly twice as many examples as the train set. To increase the number of training examples while maintaining a rigorous evaluation, we elected to perform 5-fold cross-validation on the combined data from both partitions. We used 10% of the data within each fold as a validation set, 20% as a test set and the remaining data was used as a train set. We report the micro $F_1$ score averaged across all folds.

To the best of our knowledge, we are the first to evaluate a joint NER and RE model on the 2010 i2b2/VA dataset. Therefore, we decided to compare to scores obtained by independent NER and RE systems. We note, however, that the scores of independent RE systems are not directly comparable to the scores we report in this paper. This is because RE is traditionally framed as a sentence-level classification problem. During pre-processing, each example is permutated into processed examples containing two "blinded" entities and labelled for one relation class. E.g. the example: "His PCP had recently started ciprofloxacin$_{\text{TREATMENT}}$ for a UTI$_{\text{PROBLEM}}$" becomes "His PCP had recently started @TREATMENT$ for a @PROBLEM$", where the model is trained to predict the target relation type, "Treatment is administered for medical problem" (TrAP).

This task is inherently easier than the joint setup, for two reasons: relation predictions are made on ground-truth entities, as opposed to predicted entities (which are noisy) and the model is only required to make one classification decision per pre-processed sentence. In the joint setup, a model must identify any number of relations (or the lack thereof) between all unique pairs of *predicted* entities in a given input sentence. To control for the first of these differences, we report scores from our model in two settings, once when predicted entities are used as input to the RE module, and once when ground-truth entities are used.

## 3.2 HYPERPARAMETERS

Besides batch size, learning rate and number of training epochs, we used the same hyperparameters across all experiments (see Table A.2). Similar to Devlin et al. (2018), learning rate and batch size were selected for each dataset using a minimal grid search (see See Table A.3).

---

[5] https://github.com/pgcool/TF-MTRNN/tree/master/data/CoNLL04
[6] https://www.ncbi.nlm.nih.gov/pubmed

One hyperparameter selected by hand was the choice of the pre-trained weights used to initialize the $BERT_{BASE}$ model. For general domain corpora, we found the cased $BERT_{BASE}$ weights from Devlin et al. (2018) to work well. For biomedical corpora, we used the weights from BioBERT (Lee et al., 2019), which recently demonstrated state-of-the-art performance for biomedical NER, RE and QA. Similarly, for clinical corpora we use the weights provided by Peng et al. (2019), who pre-trained $BERT_{BASE}$ on PubMed abstracts and clinical notes from MIMIC-III[7].

## 4 RESULTS

### 4.1 JOINTLY LEARNING NER AND RE

Table 1 shows our results in comparison to previously published results, grouped by the domain of the evaluated corpus. We find that on every dataset besides i2b2, our model improves NER performance, for an average improvement of ∼2%. This improvement is particularly large on the ACE04 and ACE05 corpora (3.98% and 2.41% respectively). On i2b2, our joint model performs within 0.29% of the best independent NER solution.

For relation extraction, we outperform previous methods on 2 datasets and come within ∼2% on both ACE05 and CoNLL04. In two cases, our performance improvement is substantial, with improvements of 4.59% and 10.25% on the ACE04 and ADE corpora respectively. For i2b2, our score is not directly comparable to previous systems (as discussed in section 3.1.4) but will facilitate future comparisons of joint NER and RE methods on this dataset. By comparing overall performance, we find that our approach achieves new state-of-the-art performance for 3 popular benchmark datasets (ACE04, ACE05, ADE) and comes within 0.2% for CoNLL04.

### 4.2 ABLATION ANALYSIS

To determine which training strategies and components are responsible for our models performance, we conduct an ablation analysis on the CoNLL04 corpus (Table 2). We perform five different ablations: (a) Without entity pre-training (see section 2.1), i.e. the loss function is given by equation 9. (b) Without entity embeddings, i.e. equation 3 becomes $x_i^{(RE)} = x_i^{(NER)}$. (c) Replacing the two feed-forward neural networks, $FFNN_{head}$ and $FFNN_{tail}$ with a single FFNN (see equation 4 and 5). (d) Removing $FFNN_{head}$ and $FFNN_{tail}$ entirely. (e) Without the bilinear operation, i.e. equation 7 becomes a simple linear transformation.

Removing $FFNN_{head}$ and $FFNN_{tail}$ has, by far, the largest negative impact on performance. Interestingly, however, replacing $FFNN_{head}$ and $FFNN_{tail}$ with a single FFNN has only a small negative impact. This suggests that while these layers are very important for model performance, using distinct FFNNs for the projection of head and tail entities (as opposed to the same FFNN) is relatively much less important. The next most impactful ablation was entity pre-training, suggesting that low-performance entity detection during the early stages of training is detrimental to learning (see section 2.1). Finally, we note that the importance of entity embeddings is surprising, as a previous study has found that entity embeddings did not help performance on the CoNLL04 corpus (Bekoulis et al., 2018a), although their architecture was markedly different. We conclude that each of our ablated components is necessary to achieve maximum performance.

### 4.3 ANALYSIS OF THE WORD-LEVEL ATTENTION WEIGHTS

One advantage of including a transformer-based language model is that we can easily visualize the attention weights with respect to some input. This visualization is useful, for example, in detecting model bias and locating relevant attention heads (Vig, 2019). Previous works have used such visualizations to demonstrate that specific attention heads mark syntactic dependency relations and that lower layers tend to learn more about syntax while higher layers tend to encode more semantics (Raganato & Tiedemann, 2018).

In Figure 2 we visualize the attention weights of select layers and attention heads from an instance of BERT fine-tuned within our model on the CoNLL04 corpus. We display four patterns that are easily

---

[7]https://mimic.physionet.org/

Table 1: Comparison to previously published $F_1$ scores for joint named entity recognition (NER) and relation extraction (RE). Ours (gold): our model, when gold entity labels are used as input to the RE module. Ours: full, end-to-end model (see section 2). Bold: best scores. Subscripts denote standard deviation across three runs. $\Delta$: difference to our overall score.

| | Dataset | Method | Entity | Relation | Overall | $\Delta$ |
|---|---|---|---|---|---|---|
| General | ACE04 | Miwa & Bansal (2016) | 81.80 | 48.40 | 65.10 | -5.69 |
| | | Bekoulis et al. (2018b) | 81.64 | 47.45 | 64.54 | -6.25 |
| | | Li et al. (2019) | 83.60 | 49.40 | 66.50 | -4.29 |
| | | Ours | **87.58**$_{0.2}$ | **53.99**$_{0.1}$ | **70.79** | – |
| | ACE05 | Miwa & Bansal (2016) | 83.40 | 55.60 | 69.50 | -3.42 |
| | | Zhang et al. (2017) | 83.50 | 57.50 | 70.50 | -2.42 |
| | | Li et al. (2019) | 84.80 | **60.20** | 72.50 | -0.42 |
| | | Ours | **87.21**$_{0.3}$ | 58.63$_{0.0}$ | **72.92** | – |
| | CoNLL04 | Miwa & Sasaki (2014) | 80.70 | 61.00 | 70.85 | -7.30 |
| | | Bekoulis et al. (2018b) | 83.61 | 61.95 | 72.78 | -5.37 |
| | | Li et al. (2019) | 87.80 | **68.90** | **78.35** | 0.20 |
| | | Ours | **89.46**$_{1.0}$ | 66.83$_{0.4}$ | 78.15 | – |
| Biomedical | ADE | Li et al. (2016) | 79.50 | 63.40 | 71.45 | -16.21 |
| | | Li et al. (2017) | 84.60 | 71.40 | 78.00 | -9.66 |
| | | Bekoulis et al. (2018b) | 86.73 | 75.52 | 81.13 | -6.53 |
| | | Ours | **89.56**$_{0.1}$ | **85.77**$_{0.4}$ | **87.66** | – |
| Clinical | i2b2[*] | Si et al. (2019)[**] | **89.55** | – | – | – |
| | | Peng et al. (2019) | – | 76.40 | – | – |
| | | Ours (gold) | – | 72.03$_{0.1}$ | – | – |
| | | Ours | 89.26$_{0.1}$ | 63.02$_{0.3}$ | 76.14 | – |

[*] To the best of our knowledge, there are no published joint NER and RE models that evaluate on the i2b2 2010 corpus. We compare our model to the state-of-the-art for each individual task (see section 3.1.4).

[**] We compare to the scores achieved by their $BERT_{BASE}$ model.

Table 2: Ablation experiment results on the CoNLL04 corpus. Scores are reported as a micro-averaged $F_1$ score on the validation set, averaged across three runs of 5-fold cross-validation. (a) Without entity pre-training (section 2.1). (b) Without entity embeddings (eq. 3). (c) Using a single FFNN in place of $FFNN_{head}$ and $FFNN_{tail}$ (eq. 4 and 5) (d) Without $FFNN_{head}$ and $FFNN_{tail}$ (e) Without the bilinear operation (eq. 7). Bold: best scores. Subscripts denote standard deviation across three runs. $\Delta$: difference to the full models score.

| Model | Entity | Relation | Overall | $\Delta$ |
|---|---|---|---|---|
| Full model | 86.32$_{0.2}$ | **60.29**$_{0.2}$ | **73.30** | – |
| (a) w/o Entity pre-training | 85.91$_{0.1}$ | 59.44$_{0.2}$ | 72.67 | -0.63 |
| (b) w/o Entity embeddings | 86.07$_{0.1}$ | 59.52$_{0.2}$ | 72.80 | -0.51 |
| (c) Single FFNN | 86.02$_{0.1}$ | 60.13$_{0.0}$ | 73.07 | -0.23 |
| (d) w/o Head/Tail | 83.93$_{0.2}$ | 54.67$_{0.8}$ | 69.30 | -4.00 |
| (e) w/o Bilinear | **86.35**$_{0.1}$ | 59.60$_{0.0}$ | 72.98 | -0.33 |

interpreted: paying attention to the next and previous words, paying attention to the word itself, and paying attention to the end of the sentence. These same patterns have been found in pre-trained BERT models that have not been fine-tuned on a specific, supervised task (Vig, 2019; Raganato & Tiedemann, 2018), and therefore, are retained after our fine-tuning procedure.

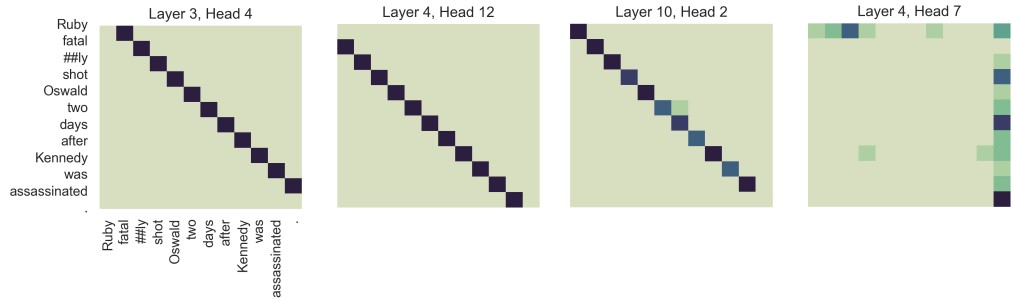

Figure 2: Visualization of the attention weights from select layers and heads of BERT after it was fine-tuned within our model on the CoNLL04 corpus. Darker squares indicate larger attention weights. Attention weights are shown for the input sentence: "Ruby fatally shot Oswald two days after Kennedy was assassinated.". The CLS and SEP tokens have been removed. Four major patterns are displayed: paying attention to the next word (first image from the left) and previous word (second from the left), paying attention to the word itself (third from the left) and the end of the sentence (fourth from the left).

To facilitate further analysis of our learned model, we make available Jupyter and Google Colaboratory notebooks on our GitHub repository[8], where users can use multiple views to explore the learned attention weights of our models. We use the BertViz library (Vig, 2019) to render the interactive, HTML-based views and to access the attention weights used to plot the heat maps.

## 5    DISCUSSION AND CONCLUSION

In this paper, we introduced an end-to-end model for entity and relation extraction. Our key contributions are: (1) No reliance on any hand-crafted features (e.g. templated questions) or external NLP tools (e.g. dependency parsers). (2) Integration of a pre-trained, transformer-based language model. (3) State-of-the-art performance on 5 datasets across 3 domains. Furthermore, our model is inherently modular. One can easily initialize the language model with pre-trained weights better suited for a domain of interest (e.g. BioBERT for biomedical corpora) or swap BERT for a comparable language model (e.g. XLNet (Yang et al., 2019)). Finally, because of (2), our model is fast to train, converging in approximately 1 hour or less on a single GPU for all datasets used in this study.

Our model out-performed previous state-of-the-art performance on ADE by the largest margin (6.53%). While exciting, we believe this corpus was particularly easy to learn. The majority of sentences (∼68%) are annotated for two entities (*drug* and *adverse effect*, and one relation (*adverse drug event*). Ostensibly, a model should be able to exploit this pattern to get near-perfect performance on the majority of sentences in the corpus. As a test, we ran our model again, this time using ground-truth entities in the RE module (as opposed to predicted entities) and found that the model very quickly reached almost perfect performance for RE on the test set (∼98%). As such, high performance on the ADE corpus is not likely to transfer to real-world scenarios involving the large-scale annotation of diverse biomedical articles.

In our experiments, we consider only intra-sentence relations. However, the multiple entities within a document generally exhibit complex, inter-sentence relations. Our model is not currently capable of extracting such inter-sentence relations and therefore our restriction to intra-sentence relations will limit its usefulness for certain downstream tasks, such as knowledge base creation. We also ignore the problem of nested entities, which are common in biomedical corpora. In the future, we would like to extend our model to handle both nested entities and inter-sentence relations. Finally, given that multilingual, pre-trained weights for BERT exist, we would also expect our model's performance to hold across multiple languages. We leave this question to future work.

---

[8]`to-add-to-camera-ready-version`

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

# A    APPENDIX

## A.1    CORPUS STATISTICS

Table A.1 lists detailed statistics for each corpus used in this study.

Table A.1: Detailed entity and relation counts for each corpus used in this study.

|  | Dataset | Entity classes (count) | Relation classes (count) |
|---|---|---|---|
| General | ACE04 | Person (12508), Organization (4405), Geographical Entities (4425), Location (614), Facility (688), Weapon (119), and Vehicle (209) | Physical (1202), Person-Social (362), Employment-Membership-Subsidiary (1591), Agent-Artifact (212), Person-Organization-Affiliation (141), Geopolitical Entity-Affiliation (517) |
|  | ACE05 | Person (20891), Organization (5627), Geographical Entities (7455), Location(1119), Facility (1461), Weapon (911), and Vehicle (919) | Physical (1612), Part-Whole (1060), Person-Social (615), Agent-Artifact (703), Employment-Membership-Subsidiary (1922), Geopolitical Entity-Affiliation (730) |
|  | CoNLL04 | Location (4765), Organization (2499), People (3918), Other (3011) | Kill (268), Live in (521), Located in (406), OrgBased in (452), Work for (401) |
| Biomedical | ADE | Drug (4979), Adverse effect (5669) | Adverse drug event (6682) |
| Clinical | i2b2 | Problem (19664), Test (13831), Treatment (14186) | PIP (2203), TeCP (504), TeRP (3053), TrAP (2617), TrCP (526), TrIP (203), TrNAP (174), TrWP (133) |

## A.2    HYPERPARAMETERS AND MODEL DETAILS

Table A.2 lists hyperparameters and model details that were held constant across all experiments. Table A.3 lists those that were specific to each evaluated corpus.

Table A.2: Hyperparameter values and model details used across all experiments.

| Hyperparameter | Value | Comment |
|---|---|---|
| Tagging scheme | BIOES | Single token entities are tagged with an S- tag, the beginning of an entity span with a B- tag, the last token of an entity span with an E- tag, and tokens inside an entity span with an I- tag. |
| Dropout rate | 0.1 | Dropout rate applied to the output of all FFNNs and the attention heads of the BERT model. |
| Entity embeddings | 128 | Output dimension of the entity embedding layer. |
| $FFNN_{head}$ / $FFNN_{tail}$ | 512 | Output dimension of the $FFNN_{head}$ and $FFNN_{tail}$ layers |
| No. layers (NER module) | 1 | Number of layers used in the FFNN of the NER module. |
| No. layers (RE module) | 2 | Number of layers used in the FFNNs of the RE module. |
| Optimizer | AdamW | Adam with fixed weight decay regulatization (Loshchilov & Hutter, 2017). |
| Gradient normalization | $\Gamma = 1$ | Rescales the gradient whenever the norm goes over some threshold $\Gamma$ (Pascanu et al., 2013). |
| Weight decay | 0.1 | L2 weight decay. |

Table A.3: Hyperparameter values specific to individual datasets. Similar to Devlin et al. (2018), a minimal grid search was performed over the values 16, 32 for batch size and 2e-5, 3e-5, and 5e-5 for learning rate.

| Dataset | No. Epochs | Batch size | Learning rate | Initial BERT weights |
|---------|-----------|-----------|---------------|---------------------|
| ACE04   | 15 | 16 | 2e-5 | BERT-Base (cased) (Devlin et al., 2018) |
| ACE05   | 15 | 32 | 3e-5 | BERT-Base (cased) |
| CoNLL04 | 10 | 16 | 3e-5 | BERT-Base (cased) |
| ADE     | 7  | 16 | 2e-5 | BioBERT (cased) (Lee et al., 2019) |
| i2b2    | 12 | 16 | 2e-5 | NCBI-BERT (uncased) (Peng et al., 2019) |

