# OpenReview forum: "End-to-end named entity recognition and relation extraction using pre-trained language models"
_ICLR.cc/2020/Conference — Reject_

### Official Review · AnonReviewer3 · 2019-10-23
**Official Blind Review #3**

**Rating:** 6

**Review:**

The paper presents an end-to-end methods for jointly training named entity recognition (NER) and relation extraction (RE). The model leverage pre-trained BERT language models, making it very fast to train. The methods is evaluated on 5 standard NER+RE datasets with good performances.

Pros:

- the paper is well written and very clear
- the proposed model has two main advantages: (1) it is very fast to train due to the use of pre-trained BERT representations and (2) it does not depends on any external NLP tool (such as dependency parser)

Cons:

- I think the main source of improvement comes from the BERT representations used as input. As proposed in the comments, this should be assessed in the paper by replacing BERT representations by non-contextual representations such as GloVE.
- Without this ablation study, the contributions of this paper are to show that using BERT representations as input (1) leads to better performances for NER+RE  and (2) makes the model faster to train. This is not really surprising...

**Experience Assessment:**

I have published in this field for several years.

**Review Assessment: Checking Correctness Of Derivations And Theory:**

N/A

**Review Assessment: Checking Correctness Of Experiments:**

I assessed the sensibility of the experiments.

**Review Assessment: Thoroughness In Paper Reading:**

N/A

---

> ### Author Response · Authors · 2019-11-05
> **Authors's Response: Official Blind Review #3**
>
> Hello,
>
> We would like to thank you for reviewing our paper. Also, thank you for your comment about the clarity of the writing, we spent a lot of effort ensuring the paper was easy to read.
>
> Regarding the suggested ablation,
>
> This comment was also made in the official blind review #2. We also responded to this suggestion in the public comment. For your convenience, we have copied our response here:
>
> In our model, BERT is more than a source of contextual word embeddings as we fine-tune all of its ~110M parameters during training. Simply replacing BERT with distributed embeddings and a character-CNN or LSTM wouldn’t allow us to determine the effect of contextualized embeddings because we would simultaneously be removing the majority of our model’s trainable parameters. Nevertheless, we performed the suggested ablation by swapping BERT for GloVe embeddings (300 dimensional) and found that NER performance dropped from 89.46% to 40.33% and RE performance fell from 66.83% to 14.44% on the test set of the ConLL04 corpus (note that we had to increase the learning rate by 10X to get the model to converge). If you were to somehow control for this drop in model capacity, say by adding in an LSTM network, the ablated model would closely match this paper [1], whom we outperform by ~3% overall on the CoNLL04 corpus. This paper is not cited in Table 1 as they report macro-averaged F1 scores, while most other papers (including the current state-of-the-art [2]) report micro-averaged F1 scores, as we did. Finally, it is well known that contextual embeddings outperform distributed embeddings on a wide range of NLP tasks, including NER [3]. The aim of our study wasn’t to compare contextual vs. distributed embeddings but on how to successfully integrate BERT into a state-of-the-art joint NER and RE architecture.
>
> Regarding your comments:
>
> “I think the main source of improvement comes from the BERT representations used as input.”
> “[...] the contributions of this paper are to show that using BERT representations as input […]”
>
> We would like to clarify that we are not simply using BERT representations as input. We are integrating BERT as part of our model architecture and fine-tuning it along with the task-specific parameters (as stated in the second to last paragraph of the introduction). For the particular problem of joint NER and RE, we found this to be critical. For example, early on in our experiments we tested using BERT as a feature extractor vs. fine-tuning the entire architecture and found that performance dropped to ~42.82% (from 78.15%) on the CoNLL04 corpus.
>
> Integrating BERT as part of our model (as opposed to simply using its embeddings as inputs) allowed us to swap recurrent architectures common in joint NER and RE models in favour of simple and shallow task-specific architectures composed of feed forward neural networks. This reduced training times while improving performance (see our response to official review #1 for more details).
>
> Again, thanks for your constructive comments!
>
> [1] https://link.springer.com/chapter/10.1007/978-3-030-15712-8_47
> [2] https://arxiv.org/abs/1905.05529
> [3] https://arxiv.org/abs/1802.05365

---

### Official Review · AnonReviewer2 · 2019-10-24
**Official Blind Review #2**

**Rating:** 3

**Review:**

The paper proposes a new joint learning algorithm that works for two tasks, NER and RE. The model is based on a pre-trained BERT model, which provides the word vectors of the input word sequence. Then it solves two tasks with two network branches: the first branch minimizes the loss for NER, and the second branch minimizes the loss for RE. The second branch uses entity labels predicted by the first branch, so joint learning may benefit both tasks.

The design of the architecture is novel, but it is also not groundbreaking. Each network branch is from known structures, but the combination is not proposed before.

The submission has evaluated the proposed algorithms on four datasets and improved SOTA performances. The ablation study justifies the design details.

The writing is generally clear.

Now critics:

Ablation study:
1. As pointed by one public comment, the ablation study should show how much improvement is from BERT vectors.

2. I'd like to see another ablation study of whether RE helps NER. If you remove the RE component, does the NER performance suffer?


Writing:
3. how are predicted labels embedded? Do you learn a vector of each tag of BIOES and then take a weighted sum of these vectors with predicted probabilities as weights?


**Experience Assessment:**

I have read many papers in this area.

**Review Assessment: Checking Correctness Of Derivations And Theory:**

I carefully checked the derivations and theory.

**Review Assessment: Checking Correctness Of Experiments:**

I assessed the sensibility of the experiments.

**Review Assessment: Thoroughness In Paper Reading:**

I read the paper thoroughly.

---

> ### Author Response · Authors · 2019-11-11
> **Authors's Response: Official Blind Review #2**
>
> Hello,
>
> Thank you for your review of our paper. We appreciate the positive assessment of the clarity of our writing.
>
> Regarding the suggested ablations,
>
> 1. For reasons outlined in our response to the public comment you reference, we do not believe this ablation (as suggested) would be meaningful. For convenience, we have copied that response here: In our model, BERT is more than a source of contextual word embeddings as we fine-tune all of its ~110M parameters during training. Simply replacing BERT with distributed embeddings and a character-CNN or LSTM wouldn’t allow us to determine the effect of contextualized embeddings because we would simultaneously be removing the majority of our model’s trainable parameters. Nevertheless, we performed the suggested ablation by swapping BERT for GloVe embeddings (300 dimensional) and found that NER performance dropped from 89.46% to 40.33% and RE performance fell from 66.83% to 14.44% on the test set of the ConLL04 corpus (note that we had to increase the learning rate by 10X to get the model to converge). If you were to somehow control for this drop in model capacity, say by adding in an LSTM network, the ablated model would closely match this paper [1], whom we outperform by ~3% overall on the CoNLL04 corpus. This paper is not cited in Table 1 as they report macro-averaged F1 scores, while most other papers (including the current state-of-the-art [2]) report micro-averaged F1 scores, as we did. Finally, it is well known that contextual embeddings outperform distributed embeddings on a wide range of NLP tasks, including NER [3]. The aim of our study wasn’t to compare contextual vs. distributed embeddings but on how to successfully integrate BERT into a state-of-the-art joint NER and RE architecture.
>
> 2. Thank you for this suggestion. We are currently performing the ablation, and will comment again once we have the results. We will be performing the same ablation as used in [2] (see section 6.2). Just note, because our manuscript is already at the page limit, we may have to place the results of this ablation in the appendix.
>
> Regarding predicted entity label embeddings,
>
> Before training, all unique entity labels (e.g. B-PER, I-PER, ... etc.) are embedded by assigning them to randomly initialized, continuous vectors of 128 dimensions (this hyperparam is mentioned in Table A.2 of the appendix). The embeddings are then updated along with the rest of the models' parameters during training. Practically speaking, this is handled for us via the embedding layer in PyTorch [4].  This is the same method used in the works we compare to ([1], [5], [6]). We have updated the text in the manuscript (under section 2) to make this more clear.
>
> Thank you again for taking the time to review our paper.
>
> [1] https://link.springer.com/chapter/10.1007/978-3-030-15712-8_47
> [2] https://arxiv.org/abs/1905.05529
> [3] https://arxiv.org/abs/1802.05365
> [4] https://pytorch.org/docs/stable/nn.html#embedding
> [5] https://www.aclweb.org/anthology/P16-1105/
> [6] https://www.sciencedirect.com/science/article/pii/S095741741830455X

---

### Official Review · AnonReviewer1 · 2019-10-24
**Official Blind Review #1**

**Rating:** 1

**Review:**

The paper proposes an end-to-end joint model for named entity recognition (NER) and relation extraction (RE), using pre-trained language models. The model is very simple, with the key is to use BERT and take NER output as input to RE. The experimental results show the model, without the need for handcrafted features, get state-of-the-art results on five datasets.

Although the paper is well written and shows good results, I would reject the paper because:
- the idea is trivial and simple. I don't think there's significant novelty here: all the components are existing and combining them seems very trivial to me.
- the good performance seems to be from BERT rather than the model's structure (table 2 suggests that). I thus think the contribution of the paper is pretty not significant.

I think the paper does not fit this conference. It is better to be presented in a Demonstration section at a *ACL conference.

**Experience Assessment:**

I have read many papers in this area.

**Review Assessment: Checking Correctness Of Derivations And Theory:**

I assessed the sensibility of the derivations and theory.

**Review Assessment: Checking Correctness Of Experiments:**

I assessed the sensibility of the experiments.

**Review Assessment: Thoroughness In Paper Reading:**

I read the paper at least twice and used my best judgement in assessing the paper.

---

> ### Author Response · Authors · 2019-11-08
> **Authors's Response: Official Blind Review #1**
>
> Hello,
>
> Thank you for reviewing our paper.
>
> Regarding your comments about the triviality of our paper,
>
> We undertook an extensive initial phase of experiments where we discovered non-trivial contributions for achieving SOTA performance and fast convergence. In particular:
>
> - To the best of our knowledge, our specific implementation of entity pre-training is novel. Our technique can be ported to other neural, multi-task setups with a strong dependence between tasks (e.g. where a model must learn to perform some task before attempting to learn one or more dependent tasks jointly). For example, we have already begun porting this scheme to a neural cross-lingual summarization project of ours.  Entity pre-training accounted for a 0.63% increase in our ablation experiment. On the CoNLL04 test set, it accounts for a 1.26% boost in performance, which is large relative to historic improvement on this corpus [1].
> - As opposed to previous models (e.g. [2], [3]) we were able to drop all recurrent architectures, which reduced training times substantially. Our work is one of the first architectures for joint NER and RE to do this. Because previous papers do not report their training times, we contacted the authors of a comparable method [4] for their training times and found that our method converged between 3-35X times faster for the ACE04, ADE, and CoNLL04 corpora (keeping in mind that we did not train on the same hardware). The large range in our estimate is because [4] trained for a wide range (60-200) of epochs.
>
> Our model serves as a strong baseline for future studies on joint NER and RE architectures and provides guidance on how to best integrate a pre-trained language model into such an architecture. For the ADE corpus in particular, we advance SOTA RE performance by >10%, which is substantially larger than improvements have been historically [5]. It is also complementary to [6] (published in ACL this year), by demonstrating similar performance without the need for templated queries, which, as pointed out in our introduction, may become a limiting factor where domain expertise is required to craft such questions (e.g., for biomedical or clinical corpora).
>
> Regarding your comments on a System Demonstration submission to ACL,
>
> Our paper is a methodological advancement, not a system, tool, or demonstration [7] and is not suitable for submission to System Demonstrations at ACL.
>
> [1] https://paperswithcode.com/sota/relation-extraction-on-conll04
> [2] https://link.springer.com/chapter/10.1007/978-3-030-15712-8_47
> [3] https://arxiv.org/abs/1804.07847
> [4] https://www.sciencedirect.com/science/article/pii/S095741741830455X
> [5] https://paperswithcode.com/sota/relation-extraction-on-ade-corpus
> [6] https://arxiv.org/abs/1905.05529
> [7] https://aclweb.org/portal/content/acl-2020-call-system-demonstrations

---

### Public Comment · ~pankaj_gupta1 · 2019-09-27
**Missing Key References and Comparisons with Most Related works**

Missing references and comparisons with the most related works:

[1] Pankaj Gupta, Hinrich Schütze, Bernt Andrassy. Table Filling Multi-Task Recurrent Neural Network for Joint Entity and Relation Extraction. In COLING-2016.
[2] Heike Adel, Hinrich Schütze. Global Normalization of Convolutional Neural Networks for Joint Entity and Relation Classification. In EMNLP-2017.

Since you have used the CoNLL04 dataset from the prior works [1, 2]; however, the quantitative comparisons with these prior works are missing even the data splits are the same. Any reason?

Originally, the train/dev/test split of CoNLL04 data was provided by [1], not [2]. Please update footnote 5. I would rather cite [1].

The data split was originally released by [1] at: https://github.com/pgcool/TF-MTRNN/tree/master/data/CoNLL04

Looking forward to your revision.

---

> ### Public Comment · ~Cantona_ViVian1 · 2019-09-27
> **Ask for citation for your own paper under every submission?**
>
> Please give constructive reviews rather than ask for citation for your own paper under every submission.

---

> > ### Public Comment · ~pankaj_gupta1 · 2019-09-28
> > **Missing Related work and QUANTITATIVE comparison**
> >
> > First of all, one of the two suggested references does  not belong to me.
> >
> > I have suggested to include a comparison with related works [1, 2], not only citing them. Even the authors used the data released by [1], they don't quantitatively compare with some other  end-to-end relation extraction systems [1, 2].
> >
> > I would appreciate comparisons of proposed techniques with the related works and acknowledge the source of data (splits) they have used.  Suggestions of including missing comparisons are also part of the constructive comments.

---

> ### Author Response · Authors · 2019-09-30
> **Author's Response: Missing Key References and Comparisons with Most Related works**
>
> Hello,
>
> We would like to thank you for leaving a comment on our paper.
>
> Regarding the comparison to [1] and [2],
>
> We do not compare to the two works you have listed because we noticed differences in the evaluation that prevent a direct comparison to our scores:
>
> - In [1], a relaxed scoring criterion is used: “For multi-word entity mention, an entity is marked correct if at least one token is tagged correctly.” In our paper, an entity is marked as correct if and only if each token of the entity is tagged correctly. The relaxed criteria of [1] will lead to both higher entity and relation F1 scores than the strict criteria we employed.
> - In [2], it appears the authors used the macro F1 score. In our work, we use the micro F1 (as stated in section 3.1.2) in order to directly compare to the works we list in Table 1.
>
> More broadly, we faced the challenge of comparing to the top systems for 5 datasets. For each dataset, we used an evaluation scheme that allowed us to directly (and fairly) compare to the largest number of end-to-end systems, while simultaneously ensuring that the current state-of-the-art systems were included. Because comparing to [1] and [2] would each require different modifications to our evaluation, and because they are not currently state-of-the-art (for the strict evaluation criteria), we elected to exclude them from our comparisons. However, we have added citations to [1] and [2] in the introduction, which will appear in future versions of the manuscript.
>
> Regarding footnote 5,
>
> We have updated the footnote to the URL you provided (https://github.com/pgcool/TF-MTRNN/tree/master/data/CoNLL04). This updated footnote will be used in future versions of the manuscript. Thank you for pointing this out.

---

### Public Comment · ~Bruno_Taillé1 · 2019-10-23
**Can you quantify the impact of using Contextualized Embeddings (BERT) ?**

It seems natural that using BERT along with any preexisting RE architecture leads to SOTA performance.

Your ablation study is a good start but lacks the most important ablation.
Could you replace BERT by a non-contextual representation such as GloVE or W2V with a character-CNN or char-LSTM ?
This would enable to compare your architecture with previous SOTA fairly and quantify the impact of BERT (which is the main explanation for better performance).

You compare with Multiturn QA (Li 2019) which does use BERT.
However it also fails to provide this ablation and it is impossible to conclude on the interest of their method.

---

> ### Author Response · Authors · 2019-10-30
> **Author's Response: Can you quantify the impact of using Contextualized Embeddings (BERT) ?**
>
> Hello,
>
> We would like to thank you for leaving a comment on our paper.
>
> Regarding the suggested ablation,
>
> In our model, BERT is more than a source of contextual word embeddings as we fine-tune all of its ~110M parameters during training. Simply replacing BERT with distributed embeddings and a character-CNN or LSTM wouldn’t allow us to determine the effect of contextualized embeddings because we would simultaneously be removing the majority of our model’s trainable parameters. Nevertheless, we performed the suggested ablation by swapping BERT for GloVe embeddings (300 dimensional) and found that NER performance dropped from 89.46% to 40.33% and RE performance fell from 66.83% to 14.44% on the test set of the ConLL04 corpus (note that we had to increase the learning rate by 10X to get the model to converge).
>
> If you were to somehow control for this drop in model capacity, say by adding in an LSTM network, the ablated model would closely match this paper [1], whom we outperform by ~3% overall on the CoNLL04 corpus. This paper is not cited in Table 1 as they report macro-averaged F1 scores, while most other papers (including the current state-of-the-art [2]) report micro-averaged F1 scores, as we did.
>
> Finally, it is well known that contextual embeddings outperform distributed embeddings on a wide range of NLP tasks, including NER [3]. The aim of our study wasn’t to compare contextual vs. distributed embeddings but on how to successfully integrate BERT into a state-of-the-art joint NER and RE architecture.
>
> Regarding the expected result of BERT improving RE,
>
> While we agree that it is unsurprising that the addition of BERT leads to state-of-the-art performance, our study still provides guidance on how to successfully integrate BERT, e.g. task-specific architectures for NER and RE, entity pre-training, hyperparameters, and the demonstration that performance holds across multiple domains simply by swapping in domain-specific BERT checkpoints. To the best of our knowledge, this is also the first work to successfully integrate BERT into a joint NER and RE model, without the need for templated queries [2].
>
> [1] https://link.springer.com/chapter/10.1007/978-3-030-15712-8_47
> [2] https://arxiv.org/abs/1905.05529
> [3] https://arxiv.org/abs/1802.05365

---

### Decision · Program_Chairs · 2019-12-19

**Decision:**

Reject

**Comment:**

This paper presents an end-to-end technique for named entity recognition, that uses pre-trained models so as to avoid long training times, and evaluates it against several baselines. The paper was reviewed by three experts working in this area. R1 recommends Reject, giving the opinion that although the paper is well-written and results are good, they feel the technique itself has little novelty and that the main reason the technique works well is using BERT. R2 recommends Weak Reject based on similar reasoning, that the approach consists of existing components (albeit combined in a novel way) and suggest some ablation experiments to isolate the source of the good performance. R3 recommends Weak Accept but feels it is "unsurprising" that BERT allows for faster training and higher accuracy. In their response, authors emphasize that the application of pretraining to named entity recognition is new, and that theirs is a methodological advance, not purely a practical one (as R1 suggests and other reviews imply). They also argue it is not possible to do a fair ablation study that removes BERT, but make an attempt. The reviewers chose to keep their scores after the response. Given the split decision, the AC also read the paper. It is clear the paper has significant merit and significant practical value, as the reviews indicate. However, given that three expert reviewers -- all of whom are NLP researchers at top institutions -- feel that the contribution of the paper is weak (in the context of the expectations of ICLR) makes it not possible for us to recommend acceptance at this time.

---

> ### Author Response · Authors · 2019-12-20
> **Thanks to the Program Chairs**
>
> We thank the program chairs for the detailed reasoning regarding the decision. We are of course disappointed with the results, especially given that all reviews deemed the paper well-written with good results, with the meta-review agreeing that the "paper has significant merit and significant practical value".
>
> We are extremely disappointed that none of our responses to either the public or reviewers' comments received a response. There was a missed opportunity for a back and forth with the "NLP researchers at top institutions" who reviewed our paper but did not respond to our clarifications and rebuttals.